# Assessing Recovery Time of Ecosystems in China: Insights into Flash Drought Impacts on Gross Primary Productivity

Mengge Lu[1,2], Huaiwei Sun[1,3,4*],Yong Yang[1], Jie Xue[5], Hongbo Ling[5], Hong Zhang[6],Wenxin Zhang[2]

[1]School of Civil and Hydraulic Engineering, Huazhong University of Science and Technology, Wuhan 430074, China.
[2]Department of Physical Geography and Ecosystem Science, Lund University, Sölvegatan 12, 22362 Lund, Sweden.
[3]College of Water Conservancy & Architectural Engineering, Shihezi University, Shihezi 832000, China.
[4]Hubei Key Laboratory of Digital River Basin Science and Technology, Huazhong University of Science and Technology, Wuhan 430074, China.
[5]State Key Laboratory of Desert and Oasis Ecology, Xinjiang Institute of Ecology and Geography, Chinese Academy of Sciences, Urumqi 830011, China.
[6]School of Engineering and Built Environment, Griffith University, Gold Coast Campus, 4222, QLD, Australia.

*Correspondence to*: Huaiwei Sun (hsun@hust.edu.cn)

**Abstract.** Recovery time, referring to the duration an ecosystem needs to return to its pre-drought condition, is a fundamental indicator of ecological resilience. Recently, flash droughts (FDs) characterized by rapid onset and development have gained increasing attention. Nevertheless, the spatiotemporal patterns of gross primary productivity (GPP) recovery time and the factors influencing it remain largely unknown. In this study, we investigate the recovery time patterns of terrestrial ecosystem in China based on GPP using a Random Forest (RF) regression model and the Shapley Additive Prediction (SHAP) method. A random forest regression model was developed for analyzing the factors influencing recovery time and establish response functions through partial correlation for typical flash drought recovery periods. The dominant driving factors of recovery time were determined by using the SHAP method. The results reveal that the average recovery time across China is approximately 37.5 days, with central and southern regions experiencing the longest durations. Post-flash drought radiation emerges as the primary environmental factor, followed by aridity index and post-flash drought temperature, particularly in semi-arid/sub-humid areas. Temperature exhibits a non-monotonic relationship with recovery time, where both excessively cold and hot conditions lead to longer recovery periods. Herbaceous vegetation recovers more rapidly than woody forests, with deciduous broadleaf forests demonstrating the shortest recovery time. This study provides valuable insights for comprehensive water resource and ecosystem management and contributes to large-scale drought monitoring efforts.

## 1 Introduction

Climate change has exacerbated drought, which has significant implications for achievement the Sustainable Development Goals (SDGs) (Lindoso et al., 2018). Among the 17 SDGs outlined in the 2030 Agenda, at least five are directly linked to drought: Goal 6 "Clean water and sanitation", Goal 11 "Sustainable cities and communities", Goal 12 "Responsible production and consumption", Goal 13 "Climate action", and Goal 15 "Life on land" (Zhang et al., 2019; Nilsson et al., 2016). Flash droughts, characterized by rapid onset and intensification, have gained increasing recognition among hydrologist and general

public globally (Yuan et al., 2023). These events significantly impact terrestrial ecosystem productivity, photosynthesis, and latent heat fluxes (Zhang et al., 2020a; Yang et al., 2023). The effects of flash droughts are not only felt during the events but also persist in their aftermath, with legacy effects post-drought (Liu et al., 2023a). Recovery time—defined as the duration required for an ecosystem to return to its pre-drought state, is a fundamental aspect of ecological resilience (Schwalm et al., 2017; Wu et al., 2017). Recovery time is related to ecological thresholds, as it may trigger a critical "tipping point" that lead to shifts into new ecosystem state (Lenton et al., 2008). With the expectation of more frequent and severe flash droughts in the future (Sreeparvathy & Srinivas, 2022), exploring post-flash drought recovery trajectories is of paramount importance (Jiao et al., 2021).

Drought recovery characteristics have been extensively observed at the ecosystem scale, typically using tree ring records, productivity or greenness measurements, and satellite data (Gazol et al., 2017; Kannenberg et al., 2019). These studies have identified varied recovery times across regions and ecosystems. Grasslands exhibit longer recovery times compared to other land covers types due to shallow-rooted plants and lower soil water retention capacity (Hao et al., 2023). Conversely, recovery in croplands is more influenced by human farming practices (Darnhofer et al., 2016). In forests, mixed forests tend to recover more quickly, whereas deciduous broadleaf forests have the longest recovery periods (He et al., 2018). Hydro-meteorological conditions also play a role, with semi-arid and semi-humid regions experiencing longer recovery times than humid and arid regions (Zhang et al., 2021). The longer recovery time in semi-arid and semi-humid regions may be related to the specific challenges these regions face, such as soil conditions, water availability, and climatic variability (Huxman et al., 2004; Zhang et al., 2021).

However, the contribution of driving factors in flash drought recovery remains unclear. Some studies indicate that background value, drought return interval, post-drought meteor-hydrological conditions, and drought attributes (such as duration, intensity) are critical in regulating recovery (Kannenberg et al., 2020). Lower background value may result in more severe damage, abnormal post-drought meteor-hydrological conditions, and longer recovery times (Fu et al., 2017). Greater drought intensity and longer duration can lead to significant ecosystem losses (Godde et al., 2019). Favorable post-drought meteor-hydrological conditions (e.g., increased precipitation and suitable temperature) improve the chance of complete recovery (Jiao et al., 2021). Plant physiological response, including changes in leaf water potential and phenology, also play a crucial role in the recovery process (Miyashita et al., 2005).

While the impacts of flash droughts on ecosystems have been well-documented, the recovery process remains underexplored. For instance, studies show that solar-induced fluorescence (SIF) and SIF yield values decline post-flash drought (Yao et al., 2022), and 95% of the gross primary production (GPP) in the Indian region responded to flash droughts with an average response time of 10-19 days (Poonia et al., 2021). However, most research focus on the immediate ecological responses to flash droughts, rather than on the recovery process (Otkin et al., 2019). Notably, a substantial contrast exists in the definition of recovery stages between flash droughts and traditional slow droughts (Wang et al., 2016). These results lead to the

conclusion that recovery is a part of the former, while the recovery phase of the latter usually occurs at the end of the event
(Qing et al., 2022). Furthermore, some studies suggest that flash drought recovery is more reliant on changes in soil moisture
or peak evapotranspiration, while traditional slow drought recovery is typically assessed using ecological or hydrological
indicators (Xu et al., 2023). For example, China has experienced frequent flash from 1980 to 2021, particularly in southwestern
and central regions (Wang et al., 2022a). Moreover, there may be more severe and frequent flash droughts in the future
(Christian et al., 2023). Research on flash drought recovery in Xiang and Wei River Basin found that most events recovered
within 28 days (Wang et al., 2023a). However, there remains a lack of comprehensive studies on flash drought recovery and
the factors influencing its spatiotemporal patterns across China.
Drought can lead to water shortages, limiting access to clean drinking water. Effective drought management is therefore crucial
for achieving SDGs. By utilizing newly available datasets and hydro-meteorological variables in China, this study assesses the
extent of post-flash drought impacts, documents recovery times, and analyzes the factors contributing to variations in
ecosystem recovery. The objectives of this study are to: (1) investigate the spatial pattern of post-flash drought recovery; (2)
identify the most critical determinants of recovery; and (3) analyze the impact of various factors on flash drought recovery
times. The following sections include Section 2, which provides a brief description of data and methods, Section 3, which
presents the results presented by novel methods applied. Then, we provide a detailed discussion in Section 4. Section 5 gives
the conclusions with some more information presented in supplementary materials.
**2 Data and methods**
**2.1 Data**
**2.1.1 Soil moisture datasets**
Daily root-zone soil moisture (SM) data for the period of 2001-2018 are obtained from Global Land Evaporation Amsterdam
Model (GLEAM) (https://www.gleam.eu/). GLEAM estimates root-zone soil moisture using a multi-layer water balance
approach. The depth of the root zone varies based on the type of land cover. For tall vegetation (e.g. trees), the depth is divided
into three layers (0-10 cm, 10-100 cm, and 100-250 cm); For low vegetation (e.g. grass), there are two layers (0-10 cm and
10-100 cm); Bare soil only has one layer (0-10 cm) (Martens et al., 2017; Miralles et al., 2011). It has been widely applied in
the identification and impact assessment of flash drought events (Zha et al., 2023). We utilized the bilinear interpolation method
to resample SM from a spatial resolution of 0.25° to 0.1°, aligning it with the accuracy of other datasets. This method is
appropriate for continuous input values, easy to implement, and generally effective in converting coarse input data into spatially
refined output (Chen et al., 2020).
**2.1.2 Hydro-meteorological datasets of affecting variables of recovery time**
We analyse the recovery time considering multiple influencing factors such as meteorological variables, drought-related
variables, and land cover (He et al., 2018). Meteorological data from the China Meteorological Forcing Dataset (CMFD),
accessible at https://westdc.westgis.ac.cn/, is utilized for the period spanning 2001 to 2018 (Yang et al., 2019). The near-
surface air temperature, downward shortwave radiation, downward longwave radiation, precipitation rate and wind speed are
used in this study. VPD is calculated based on temperature, and specific humidity using Eq. (1) - (3) (Peixoto & Oort. 1996)
(Zotarelli et al., 2020).
$SVP = 0.618\exp\left(\frac{17.27T}{T+273.73}\right)$ (1)
$AVP \approx \frac{q_s \cdot p}{\varepsilon}$ (2)
$VPD = SVP - AVP$ (3)
where SVP and AVP is saturated vapor pressure and actual vapor pressure (kPa), respectively. And $T$ is temperatures (°C), $q_s$ is
the specific humidity, $p$ is the atmospheric pressure (kPa), $\varepsilon = 6.22$ is the ratio of water vapor molecular weight to dry air weight.
Aridity index is calculated as the ratio of precipitation to potential evapotranspiration. Typically, the multi-year average of the
aridity index serves as an indicator of water availability and drought timing within a particular region (Huang et al., 2016).
Aridity index is obtained from  https://doi.org/10.6084/m9.figshare.7504448.v5 (Zomer et al., 2022). To analyze the distinct
responses of different vegetation types, we employ the MODIS dataset from the International Geosphere-Biosphere
Programme (IGBP) MCD12C1 (Friedl et al., 2002) (Figure. S1).
**2.1.3 Gross primary productivity**
Gross Primary Productivity (GPP) is widely used as an indicator for monitoring post drought photosynthesis dynamics (Gazol
et al., 2018). The FluxSat GPP dataset (Version 2), derived from Moderate Resolution Imaging Spectroradiometer (MODIS),
is calibrated using FLUXNET 2015 and OneFlux tier 1 data, and validated with independent datasets (Joiner et al., 2021).
It shows strong agreement with flux data at most sites and performs reliably across a majority of global regions (Bennett et al.,
2021). Additionally, it has been widely used in examining the impacts of extreme climate events on the terrestrial carbon cycle
(Byrne et al., 2021). The dataset provides a spatial resolution of 0.05° and a daily temporal resolution. To match the flash
drought event, daily soil moisture data were resampled to 0.1° and aggregated to pentad-mean (five-days) data. This study
chooses the growing seasons (April to October) from 2001 to 2023 as the study period.
**2.2 Method**
**2.2.1 The identification of flash drought events and recovery time**
In this study, we identify flash drought events by analysing changes in soil moisture, taking into account their rapid
intensification and duration. Evaporation demand is often used as a warning indicator for flash droughts (Rigden et al., 2020).
Because it may overestimate flash droughts (Lesinger & Tian. 2022). To identify flash drought events, the daily soil moisture
data is aggregated into pentad-mean data. These averages are then converted into percentiles based on the climatology of each
pentad period during the growing season. The identification of flash droughts should meet the following criteria: soil moisture
(SM) must decrease from above the 40th percentile to below the 20th percentile within a 5-day period, with an average rate of
decline per pentad not less than the 5th percentile. A flash drought terminates if the declining SM rises back to the 20th
percentile. The duration of a flash drought event must be at least 4 pentads (20 days) (Yuan et al., 2019, Zhang et al., 2020a).
The speed of flash drought (Ospd) is the ratio of the difference between the 40th percentile and the lowest percentile of the
onset stage to the length of onset. The frequency refers to the overall number of occurrences within a given time frame (e.g.,
per year or per decade). Severity is the accumulated soil moisture percentile deficits from the threshold of 40th. We employed
anomaly GPP to estimate post-flash drought vegetation recovery times at the pixel scale. The recovery time was defined as the
period between the point when GPP reached its maximum loss and when it returned to its pre-flash drought level(Wang et
al., 2023a)(Figure.1). To ensure data consistency and minimize noise, we first applied a smoothing process to the pentad GPP
data using a 3-pentad forward-moving window at the pixel scale. After smoothing the data, we calculate the GPP anomaly
using the following equation:
$\text{GPP anomaly} = \frac{GPP - \mu_{GPP}}{\sigma_{GPP}}$ (4)
where, $\mu_{GPP}$ and $\sigma_{GPP}$ are mean and standard deviation of the pentad time series of GPP.
The beginning of the recovery stage is identified when the post-flash drought GPP anomaly is negative and reaches its
minimum value, indicating the point of maximum GPP loss. The recovery stage concludes when the GPP anomaly returns to
a positive value, signifying that productivity has reached or exceeded its pre-drought level. However, if no flash drought event
occurs during the period of negative GPP anomaly, if the GPP anomaly is already negative before the onset of the flash drought
event, or if negative GPP anomalies only occur for one pentad, the corresponding GPP data series is excluded from the analysis
to prevent misleading results.

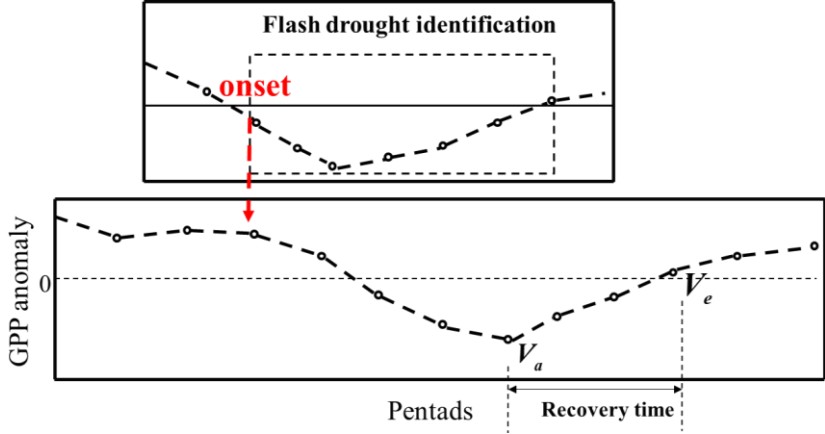


**Figure 1. The identification of recovery time.** GPP anomaly is detrended vegetation production index on a time series, 0 is
defined as the threshold of a negative anomaly. Below the dashed line represents that vegetation production is in a negative
abnormal state. We quantify recovery time as: the recovery time begins when the vegetation production loss reaches the
maximum and ends when the detrended vegetation production index is above 0.
**2.2.2 Response functions**
Partial dependence plots based on the random forest algorithm are utilized for visualizing response functions (Schwalm et al.,
2017; Sun et al., 2016). The analysis of partial dependence focuses on evaluating the marginal impact of a covariate (or
independent variable) on the response variable, while keeping other covariates constant (Liaw & Wiener. 2002). It facilitates
the exploration of insights within large datasets, particularly when random forests are primarily influenced by low-order
interactions (Martin, 2014). In addition, it is valuable tools for identifying significant features, detecting non-linear
relationships, and gaining insights into the overall behavior of a predictive model.
**2.2.3 Attribution analysis of ecosystem recovery**
In order to better understand the potential factors driving terrestrial ecosystem productivity recovery after flash droughts, we
conduct attribution analysis. We selected downward radiation (the sum of downward shortwave radiation and downward
shortwave radiation), temperature, wind speed, precipitation rate, VPD, flash drought speed (Ospd), flash drought severity
(Osev), flash drought duration (Odur), aridity index, land cover types as explanatory variables. It should be noted that these
variables are considered within the recovery period. The feature importance of random forest can only indicate the extent to
which the input variables influence the model's output, but it does not reveal how these input variables specifically impact the
model's output (Wang et al., 2022b). The Shapley Additive Prediction (SHAP) method has emerged as a valuable tool that
addresses the limitations of traditional machine learning methods (Štrumbelj&Kononenko,2014). As a result, the SHAP
method is widely utilized in attribution analysis of variables (Wang et al., 2022b; Lundberg & Lee, 2017).
$$\varphi_m(v) = \sum_{S \subseteq N \setminus \{m\}} \frac{|S|!(|N| - |S| - 1)!}{|N|!} (v(S \cup \{m\}) - v(S)) \tag{5}$$

where, $\varphi_m(v)$ represents the contribution of covariate $m$, $N$ denotes the set of all covariates, $S$ is a subset of $N$, and $v(S)$
represents the value of that subset.
We utilized a random forest model and employed these variables as predictive factors to estimate the productivity recovery
time for all study grid cells. Then, we used the SHAP value to quantify the marginal contribution of each predictive variable
and rank their relative importance based on the average absolute SHAP value.

## 3 Results

### 3.1 Characteristics of flash droughts

Figure 2 presents the frequency, duration, severity, and speed of flash droughts over China during 2001-2019. Approximately 7% of grids did not experience a flash drought event, while the remaining 93% of grids experienced at least one event. The middle and lower reaches of the Yangtze River exhibited a high frequency value with above 12 events/decade, whereas other regions mainly ranged from 0 to 9 events/decade. There is a clear spatial pattern for the duration, ranging from 0 to 20 days over China. The Southwestern and the middle and lower reaches of the Yangtze River had longer durations, exceeding 90 days (Figure. S2). In addition to the higher severity of flash droughts in the southwest region, a similar spatial pattern was observed for severity and speed. Regarding speed, areas with faster speed were primarily concentrated in the lower reaches of the Yangtze River. Overall, the middle and lower reaches of the Yangtze River and the southwestern region are considered hot spots, although the latter's speed is not rapid.

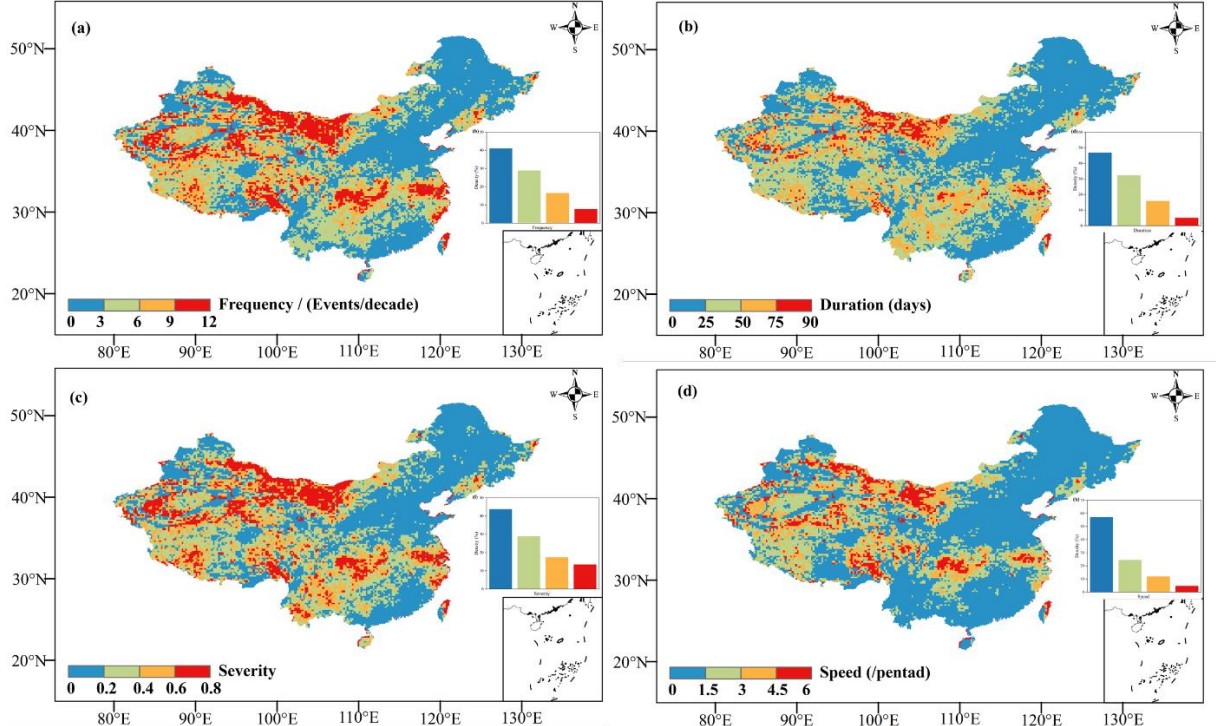

**Figure 2. Frequency (a), duration (b), severity (c), speed (d) of flash drought over China during 2001–2023.**

### 3.2 Spatial pattern of ecosystem recovery time and recovery rate

Vegetation productivity showed a clear response to flash droughts, and this response typically had a certain lag (Figure. S3). Ecosystems exhibited distinct spatial differences in recovery times to flash droughts (Figure. 3). The mean recovery time for

Chinese ecosystems was 37.5 days (7.5 pentads) calculated by GPP. Most regions were able to recover to their normal state
within 50 days. However, certain areas, such as central China and southern China, required 90 days or more to recover. In
terms of time series, there was no evident trend in the mean recovery time, with fluctuations occurring within 7.5 pentads. On
average, the recovery rate of grids in China ranged from 0 to 2 per pentad, and approximately 90% of grids had a recovery rate
of less than 1 per pentad. There is no significant trend in recovery rate over time. To further illustrate the impact and recovery
of flash droughts on different vegetation types, we calculated the recovery time and recovery rate for each type (Figure. 4).
Among the different vegetation types, DBF had a shorter recovery time and a higher recovery rate. Additionally, CRP showed
moderate recovery rates, while GRS had relatively low rates of recovery. This reflects the fact that flash droughts had a more
significant impact on GRS and resulted in greater productivity losses. By employing various recovery thresholds (80%, 90%,
100%, and 110% of the original state), we confirmed although the recovery time of some grid pixels can vary, the overall
spatial pattern of recovery time remains consistent regardless of the threshold (Figure.S4).

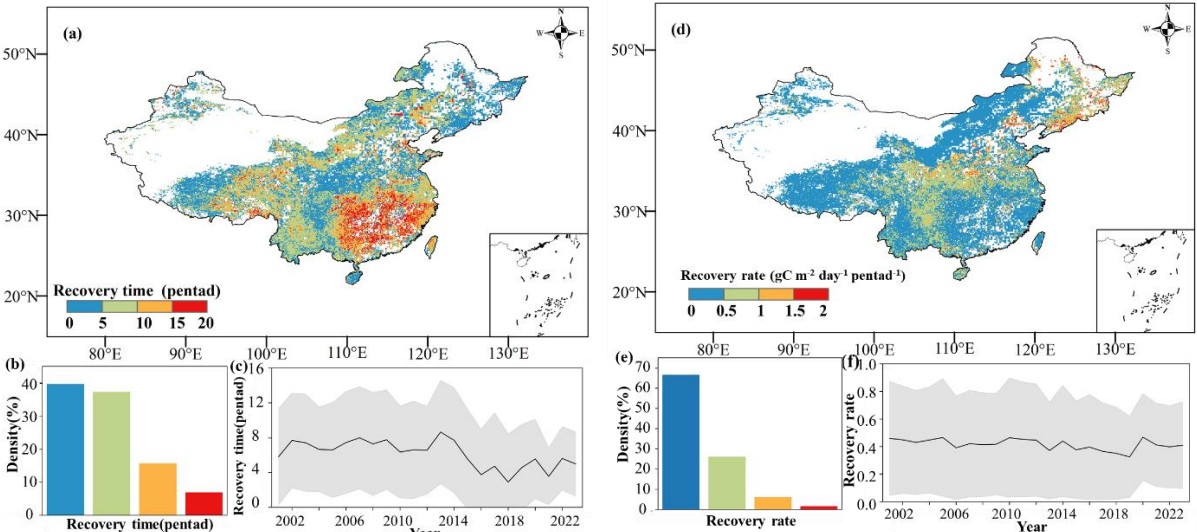


**Figure 3. Spatial pattern of recovery time (a-c) and recovery rate (d-f)**. (a) and (d) represent the recovery time (pentad)
and recovery rate (gC m$^{-2}$ day$^{-1}$ pentad$^{-1}$) calculated by using GPP data respectively. (b) and (e) represent the density of different
recovery times and recovery rate respectively, the horizontal axis represents the recovery time (pentad), recovery rate (gC m$^{-2}$
day$^{-1}$ pentad$^{-1}$) and the vertical axis is the density. Regions with sparse GPP or no droughts are masked with white.

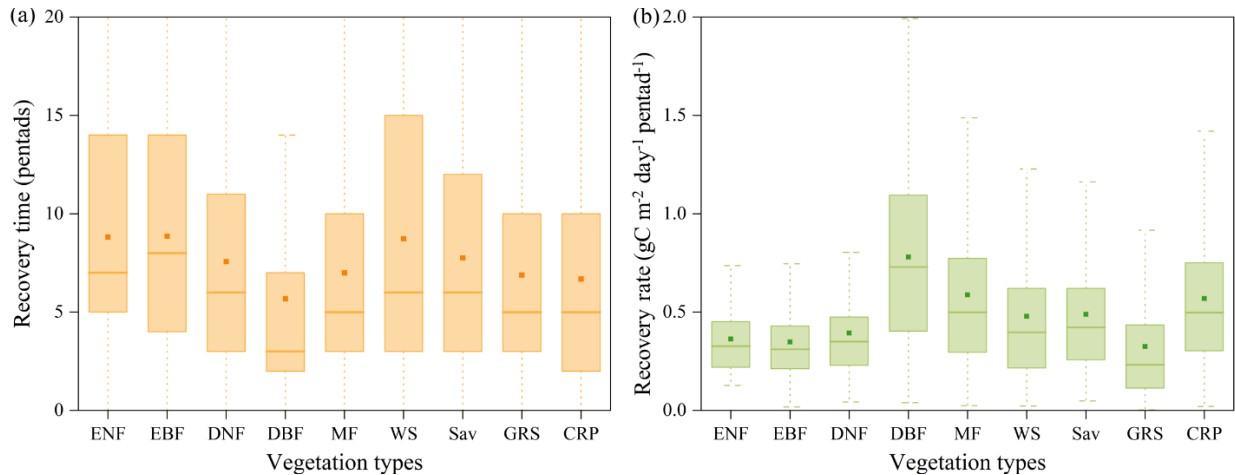


**Figure 4. The recovery time and recovery rate across different vegetation types.** The vegetation types are: ENF (evergreen coniferous forest), EBF (evergreen broad-leaved forest), DNF (deciduous coniferous forest), DBF (deciduous broad-leaved forest), MF (mixed forests), WS (closed shrubland, open shrubland, and woody savannas), SAV (savannas (temperate)), GRS (grasslands), CRP (croplands).

### 3.3 Response functions for flash drought recovery time

The random forest regression model explained 55% of the out-of-bag variance in recovery time (Figure. 5). Radiation emerged as the most influential factor impacting flash drought recovery time, with lower solar radiation conditions leading to prolonged the recovery time (Figure. 5a). Temperature did not exhibit a monotonic response in relation to recovery time. Excessively cold or overheated temperatures resulted in longer recovery times, whereas slightly higher temperatures promoted vegetation recovery (Figure. 5b). Specifically, a slight increase in temperature facilitated vegetation restoration, while higher temperatures extended the recovery time of flash droughts. This suggests that the projected rise in extreme high temperatures will further lengthen the recovery time (Li et al., 2019). In terms of flash drought characteristics, the difference in recovery time was related to the discrepancy in severity and duration, albeit to a lesser extent than speed (Figure. 5c, h & i). Recovery time increased in a stepwise manner as the duration increased. Ecosystems experiencing prolonged durations of flash droughts typically exhibit longer recovery times. In addition, semi-arid/sub-humid areas (0.2<AI<0.65) have longer recovery times (Figure. 5d). The wind speed exhibited a bimodal pattern, indicating that the recovery time was shortest when it closely aligned with the multi-year average or was 3.5 times higher than the multi-year average (Fig. 5e). Adequate precipitation following a flash drought assisted in recovery, although excessively extreme precipitation could also hinder it (Fig. 5f). Extreme vapor pressure deficit (VPD), whether high or low, prolonged the recovery time (Fig. 5g). Among different vegetation types, herbaceous vegetation recovered more rapidly than woody forests. Deciduous broadleaf forests (DBF) demonstrated the shortest recovery time (Figure. 5j).

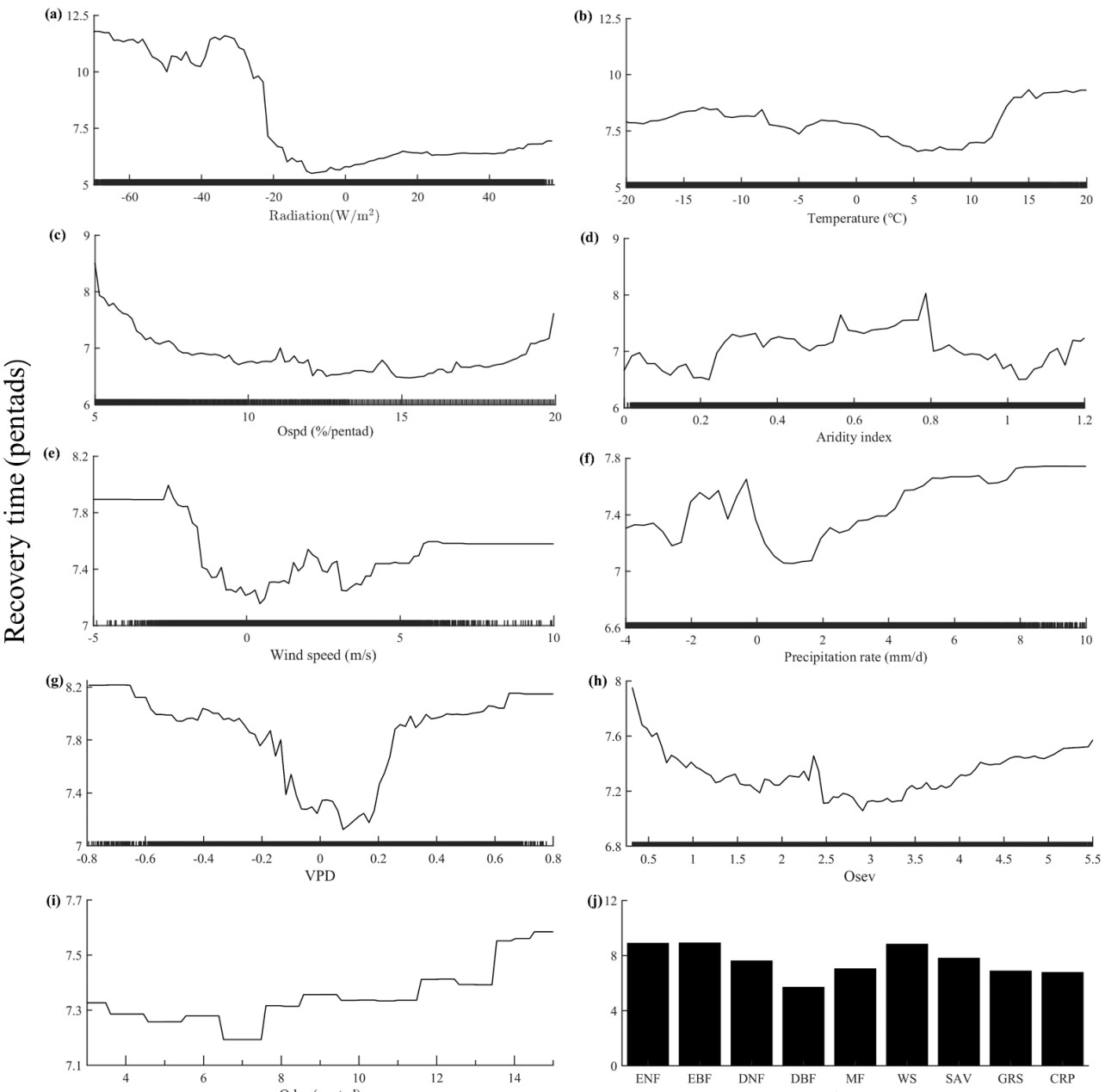

**Figure 5. Response functions for flash drought recovery time**, reflecting the response of recovery time to a single dependent variable when others are unchanged. Note difference in the y-axis scales. The covariates a to j are the deviations from the baseline. Positive (negative) indicates above (below) the average value.

**3.4 Drivers of flash drought recovery time**

We then performed an attribution analysis using SHAP method to quantify the relative importance of the considered variables. The results were consistent with the results of section 3.3. In general, radiation and aridity index were the most relevant controls of spatial variations of post-flash drought recovery time (Figure 6). Temperature was the third most impactful variable overall, primarily due to its high impact in predicting the recovery time where it has an absolute mean SHAP value of 0.62. Compared to other variables, the impact of speed and duration of flash droughts were relatively low. In addition, during the process of flash drought recovery, the losses caused by flash droughts can also affect productivity recovery. The relationship between recovery time and the attributes of flash drought (speed, severity, duration) is usually negative. That is to say, faster, more severe, and longer lasting flash droughts often have a longer recovery time. Specifically, the speed of flash droughts characteristics is one of the main controlling factors for recovery time.

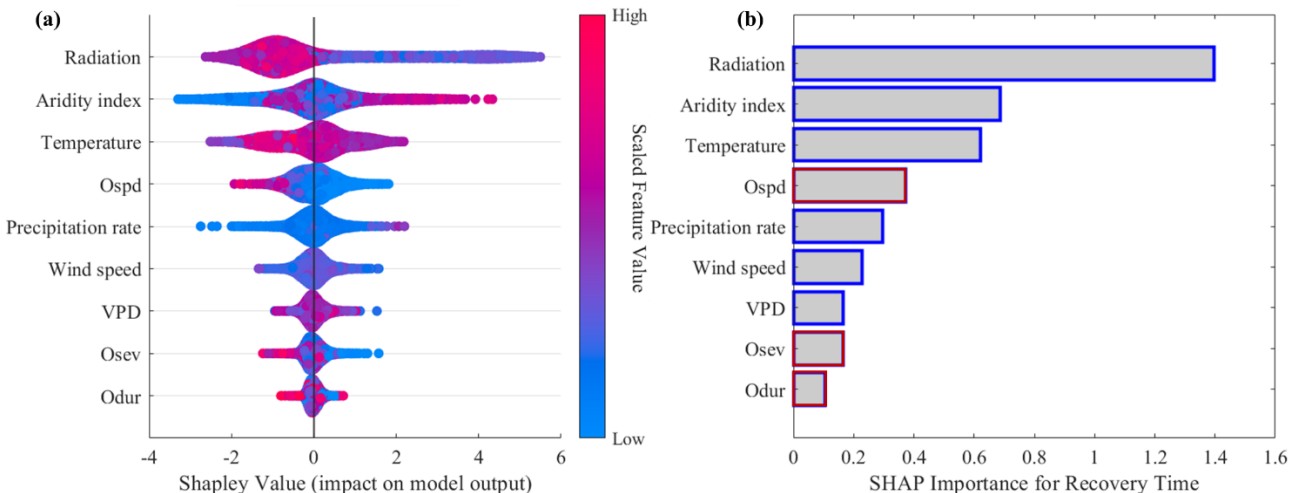

**Figure 6. Identifying drivers of patterns of post-flash drought recovery time.** (a) The summary plot of SHAP values in random forest machine learning. (b) The SHAP Importance (averaged absolute SHAP values) for recovery time. Considered drivers include flash drought characteristics (in red), post-flash drought hydro-meteorological conditions (in blue).

**4 Discussions**

**4.1 Assess flash drought recovery time based on vegetation productivity**

Given the prevalence of drought in regions over the past few decades, drought is a major natural disaster worldwide (WMO. 2021). In addition, its exposure, vulnerability, and risk are expected to further increase under future climate and socio-economic changes (Tabari & Willems. 2018; Cook et al., 2020). Flash drought is widely recognized as a sub-seasonal phenomenon that develops rapidly (Tyagi et al., 2022). Flash droughts have varying degrees of impact on the photosynthesis, productivity, and respiration of ecosystems (Mohammadi et al., 2022). Reducing drought risks and strengthening social drought resistance are

also important tasks in order to achieve SDGs by 2030 (Tabari et al., 2023). Flash droughts interact with ecological droughts, with ecological droughts potentially making ecosystems more vulnerable to flash droughts, while flash droughts can exacerbate the effects of persistent ecological droughts (Cravens et al., 2021; Xi et al., 2024). The interplay between these two types of droughts can intensify the pressure on ecosystems, complicating and prolonging the recovery process. The response frequency of Solar-Induced Fluorescence (SIF) in the China basin to flash droughts exceeds 80%, with 96.85% of the regional response occurring within 16 days (Yang et al., 2023). Previous studies have calculated the recovery time of flash drought based on changes in soil moisture, ranging from 8 to 40 days (Otkin et al., 2019). Additionally, the recovery time is generally longer in humid areas compared to arid areas. However, not all flash drought events result in a decrease in ecosystem productivity (Liu et al., 2019). For instance, a study conducted by Zhang et al. (2020b) revealed that between 2003 and 2018, 81% of flash droughts in China displayed negative normalized anomalies in GPP, while the remaining 19% of the events did not exhibit such negative anomalies. Therefore, GPP serves as a more appropriate indicator for monitoring post-drought photosynthesis-related dynamics and evaluating ecosystem recovery time (Yu et al., 2017). Based on GPP, most flash drought events in the Xiangjiang River Basin (XRB) and Weihe River Basin (WRB) recovered within 2 to 8 days. Moreover, the recovery time in the XRB, which is located in a humid area, tends to be longer (Wang et al., 2023a). It should be noted that this study only investigated the aforementioned two watersheds and did not include semi-humid/semi-arid areas. Our study revealed that the average recovery time for flash droughts in the China is approximately 37.5 days (7.5 pentads) (Figure 3).

**4.2 The factors that affect drought recovery time**

The solar radiation and aridity index were the primary factors that influence the recovery time (Figures 5 & 6, Figure S5). The recovery time was regulated by a combination of drought characteristics (drought return interval, severity, duration), post-drought hydro-meteorological conditions, and vegetation physiological characteristics (Fathi-Taperasht et al., 2022; Liu et al., 2019). Physiological responses, such as the decline rate of productivity upon exposure to flash drought also influence recovery time. Notably, there is a significant negative correlation between the decline rate and the recovery rate (Lu et al., 2024). In the case of flash droughts characterized by rapid development, the speed is one of the most important factors controlling the recovery time (Figure 6). The Yangtze River Basin experienced one of the most severe flash droughts on record during the summer of 2022, primarily driven by abnormal high temperatures and abrupt changes in precipitation (Liu et al., 2023b). The high temperatures accelerated the onset of the drought (Wang et al., 2023b). As a result, the total Gross Primary Production (GPP) loss from July to October 2022 was $26.12 \pm 16.09$ Tg C, representing a decrease of approximately 6.08% compared to the 2001-2021 average (Li et al., 2024). Ecological drought, characterized by prolonged conditions lasting months to years and resulting in long-term changes to ecosystem functions and structure (Sadiqi et al., 2022). In contrast, flash drought develops rapidly within days to weeks due to extreme weather, leading to immediate reductions in soil moisture and plant health (Yuan et al., 2023). The long-term nature of ecological drought can cause profound impacts such as reduced plant populations, increased soil erosion, and decreased biodiversity, necessitating a longer recovery period (Cravens et al., 2021). In contrast, flash droughts, while shorter in duration, cause rapid plant wilting, reduced crop yields, and soil cracking, with significant

long-term consequences for ecosystem recovery (Xi et al., 2024). These two types of droughts can interact, with ecological droughts potentially making ecosystems more susceptible to flash droughts, and flash droughts exacerbating the impacts of ongoing ecological droughts (Hacke et al., 2001; Schwalm et al., 2017). The combined effects of both types can intensify stress on ecosystems, complicating and prolonging the recovery process. Previous studies have shown that the spatial patterns of flash drought recovery were similar to those of precipitation, temperature, and radiation (Wang et al., 2023a). Increased radiation energy and precipitation post a drought can promote vegetation photosynthesis (Zhang et al., 2021). Additionally, there are regional variations in the time required for drought recovery. Generally, semi-arid and semi-humid areas took longer to recover to their pre-drought state (Figure 5). Ecosystems in these areas exhibited higher overall sensitivity to drought (Vicente et al., 2013; Yang et al., 2016). Vegetation in arid areas adapted to long-term water deficit through various physiological, anatomical, and functional mechanisms, resulting in high drought resistance (Craine et al., 2013). In humid areas, sufficient water storage helped resist drought (Liu et al., 2018; Sun et al., 2023). Vegetation also played a crucial role in regulating the recovery trajectory. The drought resistance of plants was determined by various traits such as stomatal conductance, hydraulic conductivity, and cell turgor pressure (Bartlett et al., 2016; Martínez-Vilalta et al., 2017). Grasslands and shrublands could quickly recover from drought, while forest systems require longer periods of time (Gessler et al., 2017). This may because those have relatively simple vegetation structures, shorter life cycles, and faster growth rates (Ru et al., 2023). In contrast, forest systems have more complex vegetation structures and ecological processes (Tuinenburg et al., 2022). Deep roots enhance tree tolerance to drought (McDowell et al., 2008; Nardini et al., 2016). Compared to shallow roots, deep roots have larger conduit diameters and vessel cells, resulting in higher hydraulic conductivity. During droughts, deep roots may play a critical role in water absorption, as increased root growth with soil depth could represent an adaptation to drought conditions (Germon et al., 2020), enabling rapid access to substantial water reserves stored in deeper soils (Christina et al., 2017).

## 4.3 Limitations and perspectives

We emphasized that the post-flash drought recovery trajectory of ecosystem is influenced by several factors, including post-flash drought hydrological conditions, flash drought characteristics, and the physiological characteristics of vegetation. However, we should note that in this study, the same percentile threshold (20%, 40%) was used to identify flash drought events based on empirical values from previous research findings. Further investigation should investigate how to determine region-specific thresholds and examine the sensitivity of these thresholds to flash drought recognition (Gou et al., 2022). Furthermore, it is important to consider that plant strategies for coping with flash drought can vary due to species differences (Gupta et al., 2020). There is still a need for improvement in understanding the physiological and ecological mechanisms involved in flash drought recovery. To gain a more comprehensive understanding, future research should explore the mechanism of ecosystem restoration from multiple perspectives, such as evaluating greenness and photosynthesis. Although flash droughts can lead to significant short-term disruptions, there remains a need to explore their long-term effects more comprehensively. Future research should prioritize understanding how these intense, short-term drought events might evolve into more conventional

droughts and the persistence of their impacts over time (Liu et al., 2023a). Understanding these dynamics will be crucial for
predicting and managing the carbon balance and resilience of ecosystems under changing climate conditions.
**5 Conclusions**
Effectively reducing drought risk and reducing drought exposure are crucial for achieving sustainable development goals
(SDGs) related to health and food security. This study applied a random forest regression model to analyze the factors
influencing recovery time and the response functions settled up by partial correlation for typical flash drought recovery time.
The most important environmental factor affecting recovery time is post-flash drought radiation, followed by aridity index and
post-flash drought temperature. Recovery time prolongs with lower solar radiation conditions. Semi-arid/sub-humid areas have
longer recovery time. Temperature does not exhibit a monotonic response in relation to recovery time; excessively cold or
overheated temperatures lead to longer recovery times. Herbaceous vegetation recovers more rapidly than woody forests, with
deciduous broadleaf forests demonstrating the shortest recovery time.
Our study assessed the recovery time of ecosystems to flash droughts based on GPP dataset and identified the dominant factors
of recovery time. Results show that 78% of ecosystems could recover within 0 to 50 days. However, certain areas, such as
central China and southern China, required 90 days or more to recover. The analysis of the response functions showed that
radiation emerged as the most influential factor impacting flash drought recovery time, with lower solar radiation conditions
leading to prolonged recovery time. Additionally, temperature did not exhibit a monotonic response in relation to recovery
time. In terms of flash drought characteristics, the difference in recovery time is more associated with speed than severity and
duration.
Although this study provides a good basis for further investigation of flash drought characteristics, it is important to note that
the further extension of this study may lead to more understanding of flash drought for hydrological application or worldwide
practices. It is important to determine region-specific thresholds and examine the sensitivity of these thresholds to flash drought
recognition. Furthermore, plant strategies for coping with flash drought can vary due to species differences. To gain a more
comprehensive understanding of flash drought recovery, future research should also explore the mechanism of ecosystem
restoration from multiple perspectives, such as evaluating greenness and photosynthesis.

**Author contributions**
**Mengge Lu**: Conceptualization, Methodology, Data curation, Formal analysis, Writing - original draft. **Huaiwei Sun**:
Conceptualization, Project administration, Writing - review & editing, Supervision. **Yong Yang**: Writing - review & editing.
**Jie Xue**: Writing - review & editing. **Hongbo Lin**: Writing - review & editing. **HongZhang**: Writing - review & editing.
**Wenxin Zhang**: Writing - review & editing.

**Declaration of competing interest**

The authors declare that they have no known competing financial interests or personal relationships that could have appeared
to influence the work reported in this paper.

**Data availability**

Global Land Evaporation Amsterdam Model (GLEAM) soil moisture data is available from https://www.gleam.eu/. The China
Meteorological Forcing Dataset (CMFD) can be accessed via https://westdc.westgis.ac.cn/zh-hans/data/7a35329c-c53f-4267-
aa07-e0037d913a21/. The FluxSat GPP dataset (Version 2) dataset is available from https://daac.ornl.gov. The MODIS land
cover dataset MCD12C1 is available from https://doi.org/10.24381/cds.f17050d7.

**Acknowledgements**

This study was funded by the Third Xinjiang Scientific Expedition Program (Grant No.2022xjkk0105) (H.S.). The authors
also acknowledge funding from NSFC projects (51879110,52079055, 52011530128). In addition, H.S. acknowledges funding
from a NSFC-STINT project (No. 202100-3211). Mengge Lu acknowledges China Scholarship Council (grant number:

363 202306160083).

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

Sci Data 9, 409.