# Peer review of "Assessing Recovery Time of Ecosystems in China: Insights into Flash Drought Impacts on Gross Primary Productivity"

_Hydrology and Earth System Sciences, 2024_

## Referee Comment (RC1)

This manuscript investigates how ecosystem primary productivity recovers after experiencing flash droughts using the random forest model and an explainable model. These results reveal the response time of GPP over China and its influencing factors. The topic is of significance to assess flash droughts' ecological impacts, which is probably a concern to the community of hydrologists, ecologists, and policy-makers. However, the study should clarify some comments before it is accepted by *Hydrology and Earth System Sciences*.

Major comments:

1. The abstract pointed the most novel finding is proposing a new method of a machine learning method to study the recovery of GPP to flash droughts. In my opinion, the method is not quite new and has been widely used in analyzing the interactions between soil moisture and vegetation. Whereas the recovery of GPP is less involved in previous studies, this study contributes a lot to provide a perspective on this topic.

2. The metric of GPP recovery from flash droughts used in this study may be influenced by data noises, for example, during a flash drought event that persists for 2 months, negative GPP anomalies only occur for 5 days. Such cases should be excluded in the analysis. The precondition of GPP recovery from flash droughts is that GPP has been negatively influenced by flash droughts. Besides, the terminating point of the recovery process is difficult to detect and should not be recognized at the point where the GPP anomaly is above 0, as there are many noises including whether it has experienced another drought or other extreme, the stable condition may be higher or lower than the normal conditions, etc.

3. Does the declining magnitude of GPP caused by flash droughts influence GPP's recovery time?

4. What are the hydro-meteorological conditions during the recovery stage of GPP? Is there a connection between hydro-meteorological conditions and GPP recovery?

5. The study period is a little short and the available datasets have been updated to 2023 even longer.

Minor comments:

1. L23: "response function functions"

2. L41: "productivity" should be more clear. Maybe "terrestrial ecosystem productivity" is better.

3. L48-54: The phase reviews the previous research about how vegetation recovers from droughts. It seems that they are inconsistent with the recovery of GPP in terms of GPP's response across different PFTs. Is there any explanation for it?

4. L54:55 & L303:304: Vegetation over humid regions needs more time to recover to its normal condition. As there is more water available over humid regions, why vegetation is more difficult to recover?

5. L57: What is the "background value"?

6. Fig.1 is difficult for readers to understand the metric used in this study. It is better to clarify flash drought and recovery time in Fig.1 more clearly. Perhaps authors can select a case from the observed events.

7. L185: In Fig2.b, should the red line be removed?

8. L199: There is no GPP recovery over northwestern China. Is there no response of GPP to flash droughts? As usual, vegetation is more sensitive to water availability in arid or semi-arid regions than in humid regions. Besides, is the response rate the reverse value of response time? If so, they are presenting the same results.

---

## Author Comment (AC1)

**Comment # 1**

We thank the independent reviewer for the comments. The comments are all valuable and extremely helpful for revising and improving our paper, as well as the important guiding significance to our research. We have studied comments carefully and have made corrections accordingly, which we hope meet with approval. The main corrections in the paper and the response to the reviewer's comment are as follows:

This manuscript investigates how ecosystem primary productivity recovers after experiencing flash droughts using the random forest model and an explainable model. These results reveal the response time of GPP over China and its influencing factors. The topic is of significance to assess flash droughts' ecological impacts, which is probably a concern to the community of hydrologists, ecologists, and policy-makers. However, the study should clarify some comments before it is accepted by *Hydrology and Earth System Sciences*.

**R: Thank you for your summary. We really appreciate your efforts in reviewing our manuscript. We have revised the manuscript accordingly. Our point-by-point responses are detailed below.**

Major comments:

(1) The abstract pointed the most novel finding is proposing a new method of a machine learning method to study the recovery of GPP to flash droughts. In my opinion, the method is not quite new and has been widely used in analyzing the interactions between soil moisture and vegetation. Whereas the recovery of GPP is less involved in previous studies, this study contributes a lot to provide a perspective on this topic.

**R: Thank you for your thorough review and valuable feedback. We appreciate your comments regarding the novelty of the machine learning method and its extensive use in analyzing soil moisture and vegetation interactions. While the method itself may not be entirely new, our aim was to apply it in a novel way to study the recovery of GPP (Gross Primary Production) following flash droughts.**

**We hope that this specific application can provide new insights and perspectives in this area. As you pointed out, the recovery of GPP has been less explored in previous studies, and we believe our work contributes significantly to this aspect of research.**

(2) The metric of GPP recovery from flash droughts used in this study may be influenced by data noises, for example, during a flash drought event that persists for 2 months, negative GPP anomalies only occur for 5 days. Such cases should be excluded in the analysis. The precondition of GPP recovery from flash droughts is that GPP has been negatively influenced by flash droughts. Besides, the terminating point of the recovery process is difficult to detect and should not be recognized at the point where the GPP anomaly is above 0, as there are many noises including whether it has experienced another drought or other extreme, the stable condition may be higher or lower than the normal conditions, etc.

*R*: **Thank you for your insightful comments regarding the GPP recovery metrics used in our study. We acknowledge that data noise can influence the analysis, particularly in cases where negative GPP anomalies are sporadic during a flash drought event. We have already implemented a data preprocessing step to reduce the impact of noise. We agree that only those events where GPP has been significantly impacted by the flash drought should be considered in the analysis. We also recognize the challenge in accurately identifying the termination point of the recovery process. As you pointed out, simply using the point where the GPP anomaly becomes positive might not be appropriate due to various noise factors, including subsequent droughts or other extreme events, and the potential deviation of stable conditions from the norm. To address these concerns, we plan to apply stricter criteria for selecting flash drought events and further refine our methodology for determining the recovery endpoint. We will exclude cases where the negative GPP anomalies are minimal.**

(3) Does the declining magnitude of GPP caused by flash droughts influence GPP's recovery time?

**R: Thank you for your insightful question. Indeed, the declining magnitude of GPP caused by flash droughts may affect the recovery time of GPP. This impact is influenced by various biological factors. However, the current manuscript primarily focuses on abiotic factors, specifically climatic factors and the characteristics of flash droughts. We will include a discussion on how the speed and severity of GPP's response to flash droughts can influence its recovery time.**

(4) What are the hydro-meteorological conditions during the recovery stage of GPP? Is there a connection between hydro-meteorological conditions and GPP recovery?

**R: Thank you for your insightful question regarding the hydro-meteorological conditions during the recovery stage of GPP and their connection to GPP recovery. In our study, we considered a range of hydro-meteorological conditions during the recovery stage, including radiation, temperature, drought index, wind speed, precipitation rate, and vapor pressure deficit (VPD). Our results indicate a connection between these conditions and GPP recovery. Post-flash drought radiation emerged as the primary environmental factor influencing GPP recovery, followed by the aridity index and post-flash drought temperature. This connection is particularly strong in semi-arid and sub-humid areas. We also observed that temperature has a non-monotonic relationship with recovery time, where excessively cold or overheated temperatures lead to longer recovery periods.**

(5) The study period is a little short and the available datasets have been updated to 2023 even longer.

**R: Thank you for your valuable feedback. We acknowledge that the study period may be considered short. In response to your comment, we will review and incorporate the most recent datasets updated to 2023, to extend our analysis and**

**provide a more comprehensive assessment. We appreciate your suggestion and will address it in our revised manuscript.**

Minor comments:

L23: "response function functions"

**R: Thank you for pointing out the spelling error "response function functions" in our manuscript. We will correct this mistake in the revised version.**

L41: "productivity" should be more clear. Maybe "terrestrial ecosystem productivity" is better.

**R: Thank you for your suggestion to use "terrestrial ecosystem productivity" instead of "productivity" for greater clarity. We agree with your recommendation and will make this change in the revised version of our manuscript.**

L48-54: The phase reviews the previous research about how vegetation recovers from droughts. It seems that they are inconsistent with the recovery of GPP in terms of GPP's response across different PFTs. Is there any explanation for it?

**R: Thank you for your insightful question. Firstly, vegetation recovery can be assessed using various indicators such as greenness, photosynthesis, and productivity. Lines 48-54 of our manuscript provide a summary of vegetation recovery under different indicators. Our study specifically focuses on the recovery of vegetation productivity, as measured by GPP. Then, we recognize that the recovery of GPP in different plant functional types (PFTs) may appear inconsistent compared to the broader understanding of vegetation recovery from droughts. This discrepancy could be attributed to various factors, including differences in species-specific physiological responses, variations in soil and climatic conditions, or differing methodologies used in previous studies. We will investigate this further in our revised manuscript to provide a clearer explanation for these inconsistencies and enhance the discussion around the differential recovery of GPP across PFTs. Your feedback is invaluable, and we appreciate your attention to this detail.**

L54:55 & L303:304: Vegetation over humid regions needs more time to recover to its normal condition. As there is more water available over humid regions, why vegetation is more difficult to recover?

*R*: **Thank you for highlighting this point. To clarify, the statement from the article, "When comparing hydro-meteorological conditions, semi-arid and semi-humid regions have a longer recovery time compared to humid and arid regions," indicates that semi-arid and semi-humid regions generally experience a longer period to recover. This does not necessarily mean that vegetation in humid regions faces more difficulty in recovery; rather, it suggests that the recovery dynamics in semi-arid and semi-humid regions are more prolonged. The longer recovery time in semi-arid and semi-humid regions may be related to the specific challenges these regions face, such as soil conditions, water availability, and climatic variability (Huxman et al., 2004; Zhang et al., 2021). We will make sure to clarify this distinction in the revised manuscript to better explain why semi-arid and semi-humid regions may experience longer recovery times compared to other regions. Thank you for bringing this to our attention.**

L57: What is the "background value"?

*R*: **It means pre-drought vegetation conditions. Background conditions and drought-damage magnitudes played an important role in regulating drought recovery. Specifically, lower background values and greater damage led to longer recovery times (He et al., 2018).**

1 is difficult for readers to understand the metric used in this study. It is better to clarify flash drought and recovery time in Fig.1 more clearly. Perhaps authors can select a case from the observed events.

*R*: **Thank you for your feedback. We understand that clarifying the metrics used in our study is crucial for reader comprehension. We will revise Figure 1 to provide a clearer depiction of flash droughts and recovery times. Additionally, we will include a specific case study from the observed events to illustrate these concepts**

**more effectively. This should help to better convey how flash droughts and recovery times are measured and analyzed in our study. We appreciate your suggestion and will address it in the revised manuscript.**

L185: In Fig2.b, should the red line be removed?

*R*: **Thank you for your suggestion regarding Figure 2.b. We appreciate your keen observation and agree with your recommendation to remove the red line from the figure. Upon reviewing the figure, we concur that the red line does not add value to the clarity or interpretation of the data presented. We will proceed with removing the red line in the revised version of the manuscript to enhance the overall quality and accuracy of the figure. Your feedback is invaluable in ensuring the precision of our presentation, and we are grateful for your input.**

L199: There is no GPP recovery over northwestern China. Is there no response of GPP to flash droughts? As usual, vegetation is more sensitive to water availability in arid or semi-arid regions than in humid regions. Besides, is the response rate the reverse value of response time? If so, they are presenting the same results.

*R*: **The lack of GPP recovery over northwestern China in our study is primarily due to the absence or poor quality of GPP data in that region. This limitation prevents us from assessing the response of GPP to flash droughts effectively in that specific area. Regarding your question about response rate and response time, the response rate is indeed the ratio of response magnitude to response time, rather than a simple reciprocal relationship. Given that the magnitude of GPP's loss can vary spatially, the response rate and response time are not straightforward inverses of each other. While they are related, they represent different aspects of how systems respond to drought. We appreciate your attention to this detail and will make sure to incorporate a more detailed explanation in the revised version.**

**References:**

He, B., Liu, J., Guo, L., Wu, X., Xie, X., Zhang, Y., ... & Chen, Z. (2018). Recovery of ecosystem carbon and energy fluxes from the 2003 drought in Europe and the 2012 drought in the United States. *Geophysical Research Letters*, *45*(10), 4879-4888.

Huxman, T. E., Smith, M. D., Fay, P. A., Knapp, A. K., Shaw, M. R., Loik, M. E., ... & Williams, D. G. (2004). Convergence across biomes to a common rain-use efficiency. *Nature*, 429(6992), 651-654.

Zhang, S., Yang, Y., Wu, X., Li, X., & Shi, F. (2021). Postdrought recovery time across global terrestrial ecosystems. *Journal of Geophysical Research: Biogeosciences*, 126(6), e2020JG005699.

---

## Author Comment (AC2)

**Comment # 2**

We thank the independent reviewer for the comments. The comments are all valuable and extremely helpful for revising and improving our paper, as well as the important guiding significance to our research. We have studied comments carefully and have made corrections accordingly, which we hope meet with approval. The main corrections in the paper and the response to the reviewer's comment are as follows:

This study investigates the duration required for ecosystems in China to revert to their pre-flash drought state, emphasizing the spatiotemporal patterns of recovery and the factors influencing them. Particularly notable are the findings regarding the impact of post-drought radiation, aridity index, and temperature on recovery time, especially in semi-arid and sub-humid regions. These findings hold significant implications for eco-hydrological research. However, substantial revisions are necessary before the manuscript can be considered for publication.

*R*: **Thank you for your summary. We really appreciate your efforts in reviewing our manuscript. We have revised the manuscript accordingly. Our point-by-point responses are detailed below.**

Major comments:

(1) While the writing is satisfactory and effectively conveys the scientific ideas, the paper would benefit from further polishing to enhance clarity and coherence. Specifically, the introduction should be expanded to provide more detailed information rather than merely listing literature.

*R*: **Thank you for your insightful feedback. We appreciate your suggestion regarding the clarity and coherence of the paper. We agree that expanding the introduction could provide a more comprehensive background and strengthen the paper's foundation. We will revise the introduction to include a more detailed discussion of the relevant literature and the context of the study, rather than just listing previous works. This should help to better frame the research and highlight**

its significance. Thank you for your valuable input; it will undoubtedly enhance the quality of the paper.

(2) A more in-depth discussion is needed, particularly regarding the detailed process analysis and discussion of GPP recovery from flash droughts. The manuscript currently lacks this depth. Incorporating an analysis of the unprecedented 2022 mega-drought in the Yangtze River Basin could serve as a valuable case study to enhance the discussion.

*R*: Thank you for your valuable comment. We recognize the need for a more detailed discussion in the manuscript. To address this, we will expand the discussion section to include an in-depth analysis of the 2022 mega-drought in the Yangtze River Basin. This will encompass a thorough examination of the causes of the event and its impact on Gross Primary Production (GPP) recovery. By incorporating this case study, the manuscript will provide a concrete example to better illustrate the effects of flash drought conditions on GPP dynamics. We appreciate your suggestion and will revise the manuscript accordingly to enhance its depth and relevance.

(3) The definition of ecosystem recovery focuses on changes in Gross Primary Productivity (GPP) anomalies. However, in the current results, there are flash droughts lasting more than 100 days where GPP negative anomalies occur for only 5 days. Such cases should be excluded from the analysis.

*R*: Thank you for pointing this out. We agree that the definition of ecosystem recovery should be aligned with significant and sustained changes in Gross Primary Productivity (GPP) anomalies. To address this issue, we will refine the analysis criteria to exclude cases where flash droughts last more than 100 days but only exhibit GPP negative anomalies for a brief period of 5 days or less. This adjustment will ensure that the results more accurately reflect meaningful GPP anomalies and improve the robustness of the analysis. We appreciate your suggestion and will incorporate these changes in the revised manuscript.

Specific comments:

L23: "response function functions"

**R: Thank you for pointing out the spelling error "response function functions" in our manuscript. We will correct this mistake in the revised version.**

Line 48-49: It is interesting to study to what extent these ecosystems compensate. Even in the annual carbon balance, will flash drought have a lasting impact.

**R: Thank you for your valuable feedback. While our study primarily focuses on ecosystem recovery following flash droughts, we acknowledge the importance of understanding the broader impacts, including potential long-term effects on the annual carbon balance. In response to your suggestion, we will include a discussion on this topic in the revised manuscript to provide additional context and highlight any lasting implications. We appreciate your input and will address this aspect in the discussion section accordingly.**

Line195-197: How were these vegetation classifications determined? Briefly discussing the phenological characteristics of these classifications would be helpful.

**R: Thank you for your insightful question. To analyze the distinct responses of different vegetation types, we utilized the MODIS dataset from the International Geosphere-Biosphere Programme (IGBP) MCD12C1. This dataset provides global vegetation classifications based on various land cover types and phenological characteristics. In the revised manuscript, we will include a brief discussion on the phenological characteristics of these vegetation classifications. This additional information will help clarify how the different vegetation types were classified and their relevance to the study's analysis of ecosystem responses.**

Line 231: Please standardize the manuscript by changing all instances of "figure" to "fig".

**R: Thank you for your suggestion. I will standardize the manuscript by replacing all instances of "figure" with "fig" to ensure consistency throughout the document.**

**This change will be made in the revised version of the manuscript. We appreciate your attention to detail and will incorporate this adjustment accordingly.**

Line 247-248: Please add the discussion about ecological drought.

**R: Thank you comment. We will add a discussion on ecological drought with a focus on its relevance to flash droughts and ecosystem resilience. Ecological drought and flash drought are both types of drought phenomena, but they differ significantly in terms of time scales, impact mechanisms, and recovery processes. Ecological drought, characterized by prolonged conditions lasting months to years and resulting in long-term changes to ecosystem functions and structure (Sadiqi et al., 2022). In contrast, flash drought develops rapidly within days to weeks due to extreme weather, leading to immediate reductions in soil moisture and plant health (Yuan et al., 2023). The long-term nature of ecological drought can cause profound impacts such as reduced plant populations, increased soil erosion, and decreased biodiversity, necessitating a longer recovery period (Cravens et al., 2021). In contrast, flash droughts, while shorter in duration, cause rapid plant wilting, reduced crop yields, and soil cracking, with significant long-term consequences for ecosystem recovery (Xi et al., 2024). These two types of droughts can interact, with ecological droughts potentially making ecosystems more susceptible to flash droughts, and flash droughts exacerbating the impacts of ongoing ecological droughts. The combined effects of both types can intensify stress on ecosystems, complicating and prolonging the recovery process. We appreciate your suggestion, and this addition will be integrated into the discussion section of the revised manuscript.**

Line 251-253: The manuscript should emphasize the mechanisms underlying the study's findings. Adding a discussion on the differences between grasslands and forests, particularly focusing on root depth levels, would be beneficial.

**R: Thank you for your suggestion. To enhance the manuscript, we will include a detailed discussion on the mechanisms underlying the study's findings. Specifically,**

we will address the differences between grasslands and forests, with a focus on the following aspects:

**1. Root Depth Levels:** Comparing the root depth profiles of grasslands and forests, highlighting how these differences affect their respective responses to flash droughts and their recovery mechanisms.

**2. Mechanisms of Drought Resilience:** An exploration of how root depth influences the ability of these ecosystems to withstand and recover from flash drought conditions. This will include a discussion on water uptake, soil moisture retention, and the role of root systems in mitigating drought impacts.

**3. Ecological Implications:** Discussion how variations in root depth and other physiological characteristics between grasslands and forests contribute to their recovery dynamics. This will provide a deeper understanding of the study's findings in the context of different vegetation types.

Incorporating this discussion will provide a more comprehensive view of the factors driving the observed differences in ecosystem responses and recovery. Thank you for highlighting this important aspect.

**References:**

Cravens, A. E., McEvoy, J., Zoanni, D., Crausbay, S., Ramirez, A., & Cooper, A. E. (2021). Integrating ecological impacts: perspectives on drought in the Upper Missouri Headwaters, Montana, United States. *Weather, Climate, and Society*, 13(2), 363-376.

Sadiqi, S. S. J., Hong, E. M., Nam, W. H., & Kim, T. (2022). An integrated framework for understanding ecological drought and drought resistance. *Science of The Total Environment*, 846, 157477.

Xi, X., Liang, M., & Yuan, X. (2024). Increased atmospheric water stress on gross primary productivity during flash droughts over China from 1961 to 2022. *Weather and Climate Extremes*, 44, 100667.

Yuan, X., Wang, Y., Ji, P., Wu, P., Sheffield, J., & Otkin, J. A. (2023). A global transition to flash droughts under climate change. *Science*, 380(6641), 187-191.

---

## Author Response (AR1)

**Response letter**

In this letter, we give a point-by-point response to the reviews, including all relevant changes made in the manuscript. We sincerely thank the editor and all reviewers for their valuable feedback that we have used to improve the quality of our manuscript. According to the editor and reviewers' comments, we have made extensive modifications to our manuscript and supplemented extra data to make our results convincing.

**1 Response to RC1**

We thank the independent reviewer for the comments. The comments are all valuable and extremely helpful for revising and improving our paper, as well as the important guiding significance to our research. We have studied comments carefully and have made corrections accordingly, which we hope meet with approval. The main corrections in the paper and the response to the reviewer's comment are as follows:

This manuscript investigates how ecosystem primary productivity recovers after experiencing flash droughts using the random forest model and an explainable model. These results reveal the response time of GPP over China and its influencing factors. The topic is of significance to assess flash droughts' ecological impacts, which is probably a concern to the community of hydrologists, ecologists, and policy-makers. However, the study should clarify some comments before it is accepted by *Hydrology and Earth System Sciences*.

**R: Thank you for your summary. We really appreciate your efforts in reviewing our manuscript. We have revised the manuscript accordingly. Our point-by-point responses are detailed below.**

Major comments:

(1) The abstract pointed the most novel finding is proposing a new method of a machine learning method to study the recovery of GPP to flash droughts. In my opinion, the

method is not quite new and has been widely used in analyzing the interactions between soil moisture and vegetation. Whereas the recovery of GPP is less involved in previous studies, this study contributes a lot to provide a perspective on this topic.

***R***: **Thank you for your thorough review and valuable feedback. We appreciate your comments regarding the novelty of the machine learning method and its extensive use in analyzing soil moisture and vegetation interactions. As you noted, we have already removed the claims regarding the novelty of the method. While the method itself may not be entirely new, our aim was to apply it in a novel way to study the recovery of GPP (Gross Primary Production) following flash droughts. We hope that this specific application can provide new insights and perspectives in this area. As you pointed out, the recovery of GPP has been less explored in previous studies, and we believe our work contributes to this aspect of research. The modified version of abstract is as follows:**

"Recovery time, referring to the duration an ecosystem needs to return to its pre-drought condition, is a fundamental indicator of ecological resilience. Recently, flash droughts (FDs) characterized by rapid onset and development have gained increasing attention. Nevertheless, the spatiotemporal patterns of gross primary productivity (GPP) recovery time and the factors influencing it remain largely unknown. In this study, we investigate the recovery time patterns of terrestrial ecosystem in China based on GPP using a Random Forest (RF) regression model and the Shapley Additive Prediction (SHAP) method. A random forest regression model was developed for analyzing the factors influencing recovery time and establish response functions through partial correlation for typical flash drought recovery periods. The dominant driving factors of recovery time were determined by using the SHAP method. The results reveal that the average recovery time across China is approximately 37.5 days, with central and southern regions experiencing the longest durations. Post-flash drought radiation emerges as the primary environmental factor, followed by aridity index and post-flash drought temperature, particularly in semi-arid/sub-humid areas. Temperature exhibits a non-monotonic relationship with recovery time, where both excessively cold and hot

conditions lead to longer recovery periods. Herbaceous vegetation recovers more rapidly than woody forests, with deciduous broadleaf forests demonstrating the shortest recovery time. This study provides valuable insights for comprehensive water resource and ecosystem management and contributes to large-scale drought monitoring efforts."

(2) The metric of GPP recovery from flash droughts used in this study may be influenced by data noises, for example, during a flash drought event that persists for 2 months, negative GPP anomalies only occur for 5 days. Such cases should be excluded in the analysis. The precondition of GPP recovery from flash droughts is that GPP has been negatively influenced by flash droughts. Besides, the terminating point of the recovery process is difficult to detect and should not be recognized at the point where the GPP anomaly is above 0, as there are many noises including whether it has experienced another drought or other extreme, the stable condition may be higher or lower than the normal conditions, etc.

*R*: **Thank you for your insightful comments regarding the GPP recovery metrics used in our study. We acknowledge that data noise can influence the analysis, particularly in cases where negative GPP anomalies are sporadic during a flash drought event. We have already implemented a data preprocessing step to reduce the impact of noise. Specifically, we smoothed the pentad GPP using a 3-pentad forward-moving window at the pixel scale. We agree that only those events where GPP has been significantly impacted by the flash drought should be considered in the analysis. To address this, we have revised our analysis to include a minimum duration threshold (2 pentad) for negative GPP anomalies. This adjustment ensures that only significant drought impacts on GPP are considered, thereby reducing the potential influence of short-term noise. Your concerns about using the return of GPP anomalies to positive values as the endpoint of recovery, as this may oversimplify the process, particularly when subsequent extreme events or fluctuations in baseline conditions are present. We acknowledge this complexity.**

**The determination of the terminating point for the recovery process is somehow subjective. We conducted a sensitivity analysis using 90%, 100%, and 110% recovery to the original state as the threshold, and the results showed no significant differences. Following the recommendations of current literature [Wang et al., 2023; Yang et al., 2023; Zhang et al., 2020], we used the return of GPP anomaly to positive anomalies as an indicator of recovery is a widely accepted method in studies of this kind. This approach has been validated in multiple studies and has been shown to reliably capture the recovery trend of GPP.**

**The modified version of this section is as follows (Lines 134-145):**

The recovery time was defined as the period between the point when GPP reached its maximum loss and when it returned to its pre-flash drought level (Wang et al., 2023) (Fig. 1). To ensure data consistency and minimize noise, we first applied a smoothing process to the pentad GPP data using a 3-pentad forward-moving window at the pixel scale. After smoothing the data, we calculate the GPP anomaly using the following equation:

$$\text{GPP anomaly} = \frac{GPP - \mu_{GPP}}{\sigma_{GPP}} \tag{4}$$

where, $\mu_{GPP}$ and $\sigma_{GPP}$ are mean and standard deviation of the pentad time series of GPP.

The beginning of the recovery stage is identified when the post-flash drought GPP anomaly is negative and reaches its minimum value, indicating the point of maximum GPP loss. The recovery stage concludes when the GPP anomaly returns to a positive value, signifying that productivity has reached or exceeded its pre-drought level. However, if no flash drought event occurs during the period of negative GPP anomaly, if the GPP anomaly is already negative before the onset of the flash drought event, or if negative GPP anomalies only occur for one pentad, the corresponding GPP data series is excluded from the analysis to prevent misleading results.

(3) Does the declining magnitude of GPP caused by flash droughts influence GPP's

recovery time?

**R: Thank you for your insightful question. Indeed, the declining magnitude of GPP caused by flash droughts may affect the recovery time of GPP. This impact is influenced by various biological factors. However, the current manuscript primarily focuses on abiotic factors, specifically climatic factors and the characteristics of flash droughts. We included a discussion on the declining magnitude of GPP caused influence its recovery time.**

**The modified version of this section is as follows (Lines 269-273):**

"The recovery time was regulated by a combination of drought characteristics (drought return interval, severity, duration), post-drought hydro-meteorological conditions, and vegetation physiological characteristics (Fathi-Taperasht et al., 2022; Liu et al., 2019). Physiological responses, such as the decline rate of productivity upon exposure to flash drought also influence recovery time. Notably, there is a significant negative correlation between the decline rate and the recovery rate (Lu et al., 2024)."

(4) What are the hydro-meteorological conditions during the recovery stage of GPP? Is there a connection between hydro-meteorological conditions and GPP recovery?

**R: Thank you for your insightful question regarding the hydro-meteorological conditions during the recovery stage of GPP and their connection to GPP recovery. In our study, we considered a range of hydro-meteorological conditions during the recovery stage, including radiation, temperature, drought index, wind speed, precipitation rate, and vapor pressure deficit (VPD). Our results indicate a connection between these conditions and GPP recovery. Post-flash drought radiation emerged as the primary environmental factor influencing GPP recovery, followed by the aridity index and post-flash drought temperature. This connection is particularly strong in semi-arid and sub-humid areas. We also observed that temperature has a non-monotonic relationship with recovery time, where excessively cold or overheated temperatures lead to longer recovery periods. We have clarified this point in the manuscript (Lines 160-164):**

"In order to better understand the potential factors driving terrestrial ecosystem productivity recovery after flash droughts, we conduct attribution analysis. We selected downward radiation (the sum of downward shortwave radiation and downward shortwave radiation), temperature, wind speed, precipitation rate, VPD, flash drought speed (Ospd), flash drought severity (Osev), flash drought duration (Odur), aridity index, land cover types as explanatory variables. It should be noted that these variables are considered within the recovery time period."

(5) The study period is a little short and the available datasets have been updated to 2023 even longer.

*R*: **Thank you for your valuable feedback. We acknowledge that the study period may be considered short. In response to your comment, we have incorporated the most recent datasets updated to 2023, extending our analysis for a more comprehensive assessment. The results based on this updated data are detailed in the revised manuscript.**

Minor comments:

L23: "response function functions"

*R*: **Thank you for pointing out the spelling error "response function functions" in our manuscript. We have revised it in the revised version.**

L41: "productivity" should be more clear. Maybe "terrestrial ecosystem productivity" is better.

*R*: **Thank you for your suggestion to use "terrestrial ecosystem productivity" instead of "productivity" for greater clarity. We agree with your recommendation. We have revised it.**

L48-54: The phase reviews the previous research about how vegetation recovers from droughts. It seems that they are inconsistent with the recovery of GPP in terms of GPP's response across different PFTs. Is there any explanation for it?

*R*: **Thank you for your insightful question. Firstly, vegetation recovery can be assessed using various indicators such as greenness index, photosynthesis, and vegetation productivity. Lines 48-54 of our manuscript provide a summary of vegetation recovery under different indicators. Our study specifically focuses on the recovery of vegetation productivity, as measured by GPP. Then, we recognize that the recovery of GPP in different plant functional types (PFTs) may appear inconsistent compared to the broader understanding of vegetation recovery from droughts. This discrepancy could be attributed to various factors, including differences in species-specific physiological responses, variations in soil and climatic conditions, or differing methodologies used in previous studies. Your feedback is invaluable, and we appreciate your attention to this detail.**

L54:55 & L303:304: Vegetation over humid regions needs more time to recover to its normal condition. As there is more water available over humid regions, why vegetation is more difficult to recover?

*R*: **Thank you for highlighting this point. To clarify, the statement from the article, "When comparing hydro-meteorological conditions, semi-arid and semi-humid regions have a longer recovery time compared to humid and arid regions," indicates that semi-arid and semi-humid regions generally experience a longer period to recover. This does not necessarily mean that vegetation in humid regions faces more difficulty in recovery; rather, it suggests that the recovery dynamics in semi-arid and semi-humid regions are more prolonged. The longer recovery time in semi-arid and semi-humid regions may be related to the specific challenges these regions face, such as soil conditions, water availability, and climatic variability (Huxman et al., 2004; Zhang et al., 2021). We have clarified this point in the manuscript (Lines 50-54):**

"Hydro-meteorological conditions also play a role, with semi-arid and semi-humid regions experiencing longer recovery times than humid and arid regions (Zhang et al., 2021). The longer recovery time in semi-arid and semi-humid regions may be related to the specific challenges these regions face, such as soil conditions, water availability, and climatic variability (Huxman et al., 2004; Zhang et al., 2021). "

L57: What is the "background value"?

**R: It means pre-drought vegetation conditions. Background conditions and drought-damage magnitudes played an important role in regulating drought recovery. Specifically, lower background values and greater damage led to longer recovery times (He et al., 2018).**

1 is difficult for readers to understand the metric used in this study. It is better to clarify flash drought and recovery time in Fig.1 more clearly. Perhaps authors can select a case from the observed events.

**R: Thank you for your feedback. We understand that clarifying the metrics used in our study is crucial for reader comprehension. We have revised Figure 1 to provide a clearer depiction of flash droughts and recovery times. This should help to better convey how flash droughts and recovery times are measured and analyzed in our study. We appreciate your suggestion. The modified figure is shown below:**

[Figure]

**Figure 1. The identification of flash drought and recovery time.** (a) is flash drought identification base on SM percentile. (b) is detrended vegetation production index on a time series, 0 is defined as the threshold of a negative anomaly. Below the dashed line represents that vegetation production is in a negative abnormal state. We quantify recovery time as: the recovery time begins when the vegetation production loss reaches the maximum and ends when the detrended vegetation production index is above 0.

L185: In Fig2.b, should the red line be removed?

*R*: **Thank you for your suggestion regarding Figure 2.b. We appreciate your keen observation and agree with your recommendation to remove the red line from the figure. Upon reviewing the figure, we concur that the red line does not add value to the clarity or interpretation of the data presented. We have removed the red line in the revised version of the manuscript to enhance the overall quality and accuracy of the figure. Your feedback is invaluable in ensuring the precision of our presentation, and we are grateful for your input.**

[Figure]

**Figure 2. Frequency (a), duration (b), severity (c), speed (d) of flash drought over China during 2001–2019.**

L199: There is no GPP recovery over northwestern China. Is there no response of GPP to flash droughts? As usual, vegetation is more sensitive to water availability in arid or semi-arid regions than in humid regions. Besides, is the response rate the reverse value of response time? If so, they are presenting the same results.

*R*: **The lack of GPP recovery over northwestern China in our study is primarily due to the absence or poor quality of GPP data in that region. This limitation prevents us from assessing the response of GPP to flash droughts effectively in that specific area. Regarding your question about response rate and response time, the response rate is indeed the ratio of response magnitude to response time, rather than a simple reciprocal relationship. Given that the magnitude of GPP's loss can vary spatially, the response rate and response time are not straightforward inverses of each other. While they are related, they represent different aspects of how systems respond to drought. We appreciate your attention to this detail.**

**2 Response to RC2**

We thank the independent reviewer for the comments. The comments are all valuable and extremely helpful for revising and improving our paper, as well as the important guiding significance to our research. We have studied comments carefully and have made corrections accordingly, which we hope meet with approval. The main corrections in the paper and the response to the reviewer's comment are as follows:

This study investigates the duration required for ecosystems in China to revert to their pre-flash drought state, emphasizing the spatiotemporal patterns of recovery and the factors influencing them. Particularly notable are the findings regarding the impact of post-drought radiation, aridity index, and temperature on recovery time, especially in semi-arid and sub-humid regions. These findings hold significant implications for eco-hydrological research. However, substantial revisions are necessary before the manuscript can be considered for publication.

**R: Thank you for your summary. We really appreciate your efforts in reviewing our manuscript. We have revised the manuscript accordingly. Our point-by-point responses are detailed below.**

Major comments:

(1) While the writing is satisfactory and effectively conveys the scientific ideas, the paper would benefit from further polishing to enhance clarity and coherence. Specifically, the introduction should be expanded to provide more detailed information rather than merely listing literature.

**R: Thank you for your insightful feedback. We appreciate your suggestion regarding the clarity and coherence of the paper. We agree that expanding the introduction could provide a more comprehensive background and strengthen the paper's foundation. We agree with the comment and re-wrote the sentence in the revised manuscript as the following:**

"Climate change has exacerbated drought, which has significant implications for

achievement the Sustainable Development Goals (SDGs) (Lindoso et al., 2018). Among the 17 SDGs outlined in the 2030 Agenda, at least five are directly linked to drought: Goal 6 "Clean water and sanitation", Goal 11 "Sustainable cities and communities", Goal 12 "Responsible production and consumption", Goal 13 "Climate action", and Goal 15 "Life on land" (Zhang et al., 2019; Nilsson et al., 2016). Flash droughts, characterized by rapid onset and intensification, have gained increasing recognition among hydrologist and general public globally (Yuan et al., 2023). These events significantly impact terrestrial ecosystem productivity, photosynthesis, and latent heat fluxes (Zhang et al., 2020a; Yang et al., 2023). The effects of flash droughts are not only felt during the events but also persist in their aftermath, with legacy effects post-drought (Liu et al., 2023). Recovery time—defined as the duration required for an ecosystem to return to its pre-drought state, is a fundamental aspect of ecological resilience (Schwalm et al., 2017; Wu et al., 2017). Recovery time is related to ecological thresholds, as it may trigger a critical "tipping point" that lead to shifts into new ecosystem state (Lenton et al., 2008). With the expectation of more frequent and severe flash droughts in the future (Sreeparvathy & Srinivas, 2022), exploring post-flash drought recovery trajectories is of paramount importance (Jiao et al., 2021).

Drought recovery characteristics have been extensively observed at the ecosystem scale, typically using tree ring records, productivity or greenness measurements, and satellite data (Gazol et al., 2017; Kannenberg et al., 2019). These studies have identified varied recovery times across regions and ecosystems. Grasslands exhibit longer recovery times compared to other land covers types due to shallow-rooted plants and lower soil water retention capacity (Hao et al., 2023). Conversely, recovery in croplands is more influenced by human farming practices (Darnhofer et al., 2016). In forests, mixed forests tend to recover more quickly, whereas deciduous broadleaf forests have the longest recovery periods (He et al., 2018). Hydro-meteorological conditions also play a role, with semi-arid and semi-humid regions experiencing longer recovery times than humid and arid regions (Zhang et al., 2021).

However, the contribution of driving factors in flash drought recovery remains unclear. Some studies indicate that background value, drought return interval, postdrought meteor-hydrological conditions, and drought attributes (such as duration, intensity) are critical in regulating recovery (Kannenberg et al., 2020). Lower background value may result in more severe damage, abnormal post-drought meteor-hydrological conditions, and longer recovery times (Fu et al., 2017). Greater drought intensity and longer duration can lead to significant ecosystem losses (Godde et al., 2019). Favorable post-drought meteor-hydrological conditions (e.g., increased precipitation and suitable temperature) improve the chance of complete recovery (Jiao et al., 2021). Plant physiological response, including changes in leaf water potential and phenology, also play a crucial role in the recovery process (Miyashita et al., 2005).

While the impacts of flash droughts on ecosystems have been well-documented, the recovery process remains underexplored. For instance, studies show that solar-induced fluorescence (SIF) and SIF yield values decline post-flash drought (Yao et al., 2022), and 95% of the gross primary production (GPP) in the Indian region responded to flash droughts with an average response time of 10-19 days (Poonia et al., 2021). However, most research focus on the immediate ecological responses to flash droughts, rather than on the recovery process (Otkin et al., 2019). Notably, a substantial contrast exists in the definition of recovery stages between flash droughts and traditional slow droughts (Wang et al., 2016). These results lead to the conclusion that recovery is a part of the former, while the recovery phase of the latter usually occurs at the end of the event (Qing et al., 2022). Furthermore, some studies suggest that flash drought recovery is more reliant on changes in soil moisture or peak evapotranspiration, while traditional slow drought recovery is typically assessed using ecological or hydrological indicators (Xu et al., 2023). For example, China has experienced frequent flash from 1980 to 2021, particularly in southwestern and central regions (Wang et al., 2022a). Moreover, there may be more severe and frequent flash droughts in the future (Christian et al., 2023). Research on flash drought recovery in Xiang and Wei River Basin found that most events recovered within 28 days (Wang et al., 2023). However, there remains a lack of comprehensive studies on flash drought recovery and the factors influencing its spatiotemporal patterns across China.

Drought can lead to water shortages, limiting access to clean drinking water.

Effective drought management is therefore crucial for achieving SDGs. By utilizing newly available datasets and hydro-meteorological variables in China, this study assesses the extent of post-flash drought impacts, documents recovery times, and analyzes the factors contributing to variations in ecosystem recovery. The objectives of this study are to: (1) investigate the spatial pattern of post-flash drought recovery; (2) identify the most critical determinants of recovery; and (3) analyze the impact of various factors on flash drought recovery times. The following sections include Section 2, which provides a brief description of data and methods, Section 3, which presents the results presented by novel methods applied. Then, we provide a detailed discussion in Section 4. Section 5 gives the conclusions with some more information presented in supplementary materials."

**Thank you for your valuable input; it will undoubtedly enhance the quality of the paper.**

(2) A more in-depth discussion is needed, particularly regarding the detailed process analysis and discussion of GPP recovery from flash droughts. The manuscript currently lacks this depth. Incorporating an analysis of the unprecedented 2022 mega-drought in the Yangtze River Basin could serve as a valuable case study to enhance the discussion.

*R*: **Thank you for your valuable comment. We agree that the discussion would benefit from a more in-depth analysis. To address this, we have expanded the discussion section to include an in-depth analysis of the 2022 mega-drought in the Yangtze River Basin, which provides a thorough examination of the event's causes and its impact on Gross Primary Production (GPP) recovery. Incorporating this case study offers a concrete example to better illustrate the effects of flash drought conditions on GPP dynamics. We appreciate your suggestion and have revised the manuscript accordingly to enhance its depth and relevance. The sentence in the revised manuscript as the following (Lines 275-279):**

"The Yangtze River Basin experienced one of the most severe flash droughts on record

during the summer of 2022, primarily driven by abnormal high temperatures and abrupt changes in precipitation (Liu et al., 2023b). The high temperatures accelerated the onset of the drought (Wang et al., 2023b). As a result, the total Gross Primary Production (GPP) loss from July to October 2022 was $26.12 \pm 16.09$ Tg C, representing a decrease of approximately 6.08% compared to the 2001-2021 average (Li et al., 2024)."

(3) The definition of ecosystem recovery focuses on changes in Gross Primary Productivity (GPP) anomalies. However, in the current results, there are flash droughts lasting more than 100 days where GPP negative anomalies occur for only 5 days. Such cases should be excluded from the analysis.

**R: Thank you for your insightful comments regarding the GPP recovery metrics used in our study. We acknowledge that data noise can influence the analysis, particularly in cases where negative GPP anomalies are sporadic during a flash drought event. We have already implemented a data preprocessing step to reduce the impact of noise. Specifically, we smoothed the pentad GPP using a 3-pentad forward-moving window at the pixel scale. We agree that only those events where GPP has been significantly impacted by the flash drought should be considered in the analysis. To address this, we have revised our analysis to include a minimum duration threshold (2 pentad) for negative GPP anomalies. This adjustment ensures that only significant drought impacts on GPP are considered, thereby reducing the potential influence of short-term noise. We appreciate your suggestion and have incorporated these changes in the revised manuscript.**

Specific comments:

L23: "response function functions"

**R: Thank you for pointing out the spelling error "response function functions" in our manuscript. We have corrected this mistake in the revised version.**

Line 48-49: It is interesting to study to what extent these ecosystems compensate. Even in the annual carbon balance, will flash drought have a lasting impact.

**R: Thank you for your valuable feedback. While our study primarily focuses on ecosystem recovery following flash droughts, we acknowledge the importance of understanding the broader impacts, including potential long-term effects on the annual carbon balance. In general, traditional droughts have a lasting impact on the annual carbon balance of ecosystems. Droughts reduce vegetation productivity, which can potentially lead to a decrease in annual carbon stocks. However, ecosystems may exhibit compensatory mechanisms; recovery after the drought can mitigate carbon loss, especially under favorable conditions following the drought. Currently, there is no clear conclusion on whether flash drought events have lasting impacts. Moreover, flash droughts fade away under the effect of accumulated water deficits: the persistence and transition to conventional drought. The sentence in the revised manuscript as the following (Lines 315-319):**

"While flash droughts can cause significant short-term disruptions, there is a need for a more comprehensive exploration of their long-term effects. Future research should focus on understanding how these intense, short-term drought events might transition into more conventional droughts and how their impacts persist over time (Liu et al., 2023a). Gaining insights into these dynamics will be essential for predicting and managing ecosystem carbon balance and resilience in the face of changing climate conditions."

Line195-197: How were these vegetation classifications determined? Briefly discussing the phenological characteristics of these classifications would be helpful.

**R: Thank you for your insightful question. To analyze the distinct responses of different vegetation types, we utilized the MODIS dataset from the International Geosphere-Biosphere Programme (IGBP) MCD12C1. This dataset provides global vegetation classifications based on various land cover types and phenological characteristics. In the revised manuscript, we will include a brief discussion on the phenological characteristics of these vegetation classifications.**

**This additional information will help clarify how the different vegetation types were classified and their relevance to the study's analysis of ecosystem responses.**

Line 231: Please standardize the manuscript by changing all instances of "figure" to "fig".

*R*: **Thank you for your suggestion. We have standardized the manuscript by replacing all instances of "figure" with "fig" to ensure consistency throughout the document. We appreciate your attention to detail and will incorporate this adjustment accordingly.**

Line 247-248: Please add the discussion about ecological drought.

*R*: **Thank you comment. We will add a discussion on ecological drought with a focus on its relevance to flash droughts and ecosystem resilience. Ecological drought and flash drought are both types of drought phenomena, but they differ significantly in terms of time scales, impact mechanisms, and recovery processes. We appreciate your suggestion. The add sentence in the revised manuscript as the following (Lines 279-288):**

"Ecological drought, characterized by prolonged conditions lasting months to years and resulting in long-term changes to ecosystem functions and structure (Sadiqi et al., 2022). In contrast, flash drought develops rapidly within days to weeks due to extreme weather, leading to immediate reductions in soil moisture and plant health (Yuan et al., 2023). The long-term nature of ecological drought can cause profound impacts such as reduced plant populations, increased soil erosion, and decreased biodiversity, necessitating a longer recovery period (Cravens et al., 2021). In contrast, flash droughts, while shorter in duration, cause rapid plant wilting, reduced crop yields, and soil cracking, with significant long-term consequences for ecosystem recovery (Xi et al., 2024). These two types of droughts can interact, with ecological droughts potentially making ecosystems more susceptible to flash droughts, and flash droughts exacerbating

the impacts of ongoing ecological droughts (Hacke et al., 2001; Schwalm et al., 2017). The combined effects of both types can intensify stress on ecosystems, complicating and prolonging the recovery process."

Line 251-253: The manuscript should emphasize the mechanisms underlying the study's findings. Adding a discussion on the differences between grasslands and forests, particularly focusing on root depth levels, would be beneficial.

*R*: **Thank you for your suggestion. To enhance the manuscript, we have added a detailed discussion on the mechanisms underlying the study's findings. Specifically, we have addressed the differences between grasslands and forests, with a focus on root depth levels. The sentence in the revised manuscript as the following (Lines 295-305):**

[revised manuscript text omitted]

Yuan, X., Wang, Y., Ji, P., Wu, P., Sheffield, J., & Otkin, J. A. 2023. A global transition to flash droughts under climate change. *Science*, 380(6641), 187-191.

---

## Author Response (AR2)

**Comment # 1**

We thank the independent reviewer for the comments. The comments are all valuable and extremely helpful for revising and improving our paper, as well as the important guiding significance to our research. We have studied comments carefully and have made corrections accordingly, which we hope meet with approval. The main corrections in the paper and the response to the reviewer's comment are as follows:

1. I suggest refining the title to make it clearer and more concise.

**R: Thank you for your suggestion. We really appreciate your efforts in reviewing our manuscript. We have revised it as:**

**"Assessing Recovery Time of Ecosystems in China: Insights into Flash Drought Impacts on Gross Primary Productivity"**

2. Please include a sensitivity analysis that uses different thresholds as recovery conditions.

**R: We have conducted a sensitivity analysis using 80%, 90%, 100%, and 110% recovery to the original state as the threshold, and the results showed no significant differences. Following the recommendations of current literature [Wang et al., 2023; Yang et al., 2023; Zhang et al., 2020a], we used the return of GPP anomaly to positive anomalies as an indicator of recovery is a widely accepted method in studies of this kind. This approach has been validated in multiple studies and has been shown to reliably capture the recovery trend of GPP. The recovery time to return to the original state at recovery thresholds of 80%, 90%, 100%, and 110% is shown in the figure below (also can see in Supplementary materials Figure S4). The sentence in the revised manuscript as the following (Lines 199-201):**

**"By employing various recovery thresholds (80%, 90%, 100%, and 110% of the original state), we confirmed although the recovery time of some grid pixels can vary, the overall spatial pattern of recovery time remains consistent regardless of the threshold (Fig.S4)."**

[Figure]

**Figure S4. Recovery time to original state at 80%(a), 90%(b), 100%(c), and 110%(d) recovery thresholds. Box plots of recovery time under different recovery thresholds (e).** The box plots display the maximum, upper quartile, median, lower quartile, and minimum values of the recovery time distribution.

3. In Figure 1, please remove any cases not included in the analysis of GPP recovery.

*R*: **Thank you for your suggestion. We have updated Figure 1 to remove any cases not included in the GPP recovery analysis, ensuring alignment with the analysis scope. The section on flash drought identification is described in detail using textual explanations (Lines 129-132):**

**"The identification of flash droughts should meet the following criteria: soil moisture (SM) must decrease from above the 40th percentile to below the 20th**

**percentile within a 5-day period, with an average rate of decline per pentad not less than the 5th percentile. A flash drought terminates if the declining SM rises back to the 20th percentile. The duration of a flash drought event must be at least 4 pentads (20 days) (Yuan et al., 2019, Zhang et al., 2020a)."**

**The updated Figure 1 is shown below:**

[Figure]

**Figure 1. The identification of recovery time.** GPP anomaly is detrended vegetation production index on a time series, 0 is defined as the threshold of a negative anomaly. Below the dashed line represents that vegetation production is in a negative abnormal state. We quantify recovery time as: the recovery time begins when the vegetation production loss reaches the maximum and ends when the detrended vegetation production index is above 0.

**References:**

Wang, H., Zhu, Q., Wang, Y., et al., 2023. Spatio-temporal characteristics and driving factors of flash drought recovery: From the perspective of soil moisture and GPP changes. *Weather and Climate Extremes*. 42: 100605.

Yuan, X., Wang, L., Wu, P., Ji, P., Sheffield, J., Zhang, M., 2019. Anthropogenic shift towards higher risk of flash drought over China. *Nat. Commun.* 10 (1).

Yang, L., Wang, W., Wei, J., 2023. Assessing the response of vegetation photosynthesis to flash drought events based on a new identification framework. *Agricultural and Forest Meteorology*. 339: 109545.

Zhang, M., Yuan, X., 2020a. Rapid reduction in ecosystem productivity caused by flash droughts based on decade-long FLUXNET observations. *Hydrology and Earth System Sciences*. 24(11): 5579-5593.